# Asymmetric synthesis of *P*-stereogenic phosphindane oxides via kinetic resolution and their biological activity

Long Yin[1,3], Jiajia Li[1,3], Changhui Wu [2,3], Haoran Zhang[1], Wenchao Zhao[1], Zhiyuan Fan[1], Mengxuan Liu[1], Siqi Zhang[1], Mengzhe Guo [1] ✉, Xiaowei Dou [2] ✉ & Dong Guo [1] ✉

The importance of *P*-stereogenic heterocycles has been widely recognized with their extensive use as privileged chiral ligands and bioactive compounds. The catalytic asymmetric synthesis of *P*-stereogenic phosphindane derivatives, however, remains a challenging task. Herein, we report a catalytic kinetic resolution of phosphindole oxides via rhodium-catalyzed diastereo- and enantioselective conjugate addition to access enantiopure *P*-stereogenic phosphindane and phosphindole derivatives. This kinetic resolution method features high efficiency (s factor up to >1057), excellent stereoselectivities (all >20:1 dr, up to >99% *ee*), and a broad substrate scope. The obtained chiral phosphindane oxides exhibit promising therapeutic efficacy in autosomal dominant polycystic kidney disease (ADPKD), and compound **3az** is found to significantly inhibit renal cyst growth both in vitro and in vivo, thus ushering in a promising scaffold for ADPKD drug discovery. This study will not only advance efforts towards the asymmetric synthesis of challenging *P*-stereogenic heterocycles, but also surely inspire further development of *P*-stereogenic entities for bioactive small-molecule discovery.

*P*-Stereogenic molecules are widely used as chiral ligands and organocatalysts in organic synthesis[1–4], and they are also important structural motifs in pharmaceuticals and biologically active compounds[5–7]. As such, developing efficient methods for asymmetric synthesis of *P*-stereogenic molecules is an important endeavor in synthetic chemistry[8–10]. Along with the remarkable advances in catalytic asymmetric synthesis of acyclic *P*-stereogenic molecules[8–21], several catalytic asymmetric strategies such as intramolecular desymmetrization[22–27] and dynamic kinetic cross-coupling[28,29] have been successfully developed to generate cyclic *P*-stereogenic heterocycles. However, the established methods are generally restricted to the construction of 6-memebered or larger *P*-stereogenic heterocycles[22–29], and the catalytic asymmetric synthesis of 5-membered *P*-stereogenic heterocycles still remains underdeveloped (Fig. 1a). In this regard, catalytic

asymmetric synthesis of *P*-stereogenic phosphindanes, which are privileged scaffolds in chiral ligands as exemplified by the powerful BIBOP/DIME-type ligands, DuanPhos, and BeePhos family (Fig. 1b)[3,30–32], remains a formidable task. Currently, access to enantio-enriched *P*-stereogenic phosphindanes primarily relies on optical resolution of the racemates using resolving agents or chiral auxiliary-assisted synthesis[33,34]. Efforts have been devoted to realizing catalytic asymmetric synthesis of this highly useful chiral scaffold, but construction of the benzo-fused 5-membered-ring can pose huge challenges for controlling the enantioselectivity even though asymmetric synthesis of *P*-stereogenic phospholanes[35] and dibenzophospholes[36,37] has been realized. Glueck achieved a highest 70% *ee* in the palladium-catalyzed enantioselective intramolecular cyclization of functionalized secondary phosphines or their borane adducts[38]. Recently, Yang

[1]Jiangsu Key Laboratory of New Drug Research and Clinical Pharmacy, Xuzhou Medical University, Xuzhou, China. [2]Department of Chemistry, School of Science, China Pharmaceutical University, Nanjing, China. [3]These authors contributed equally: Long Yin, Jiajia Li, Changhui Wu. ✉e-mail: guomengzhe@xzhmu.edu.cn; dxw@cpu.edu.cn; guo@xzhmu.edu.cn

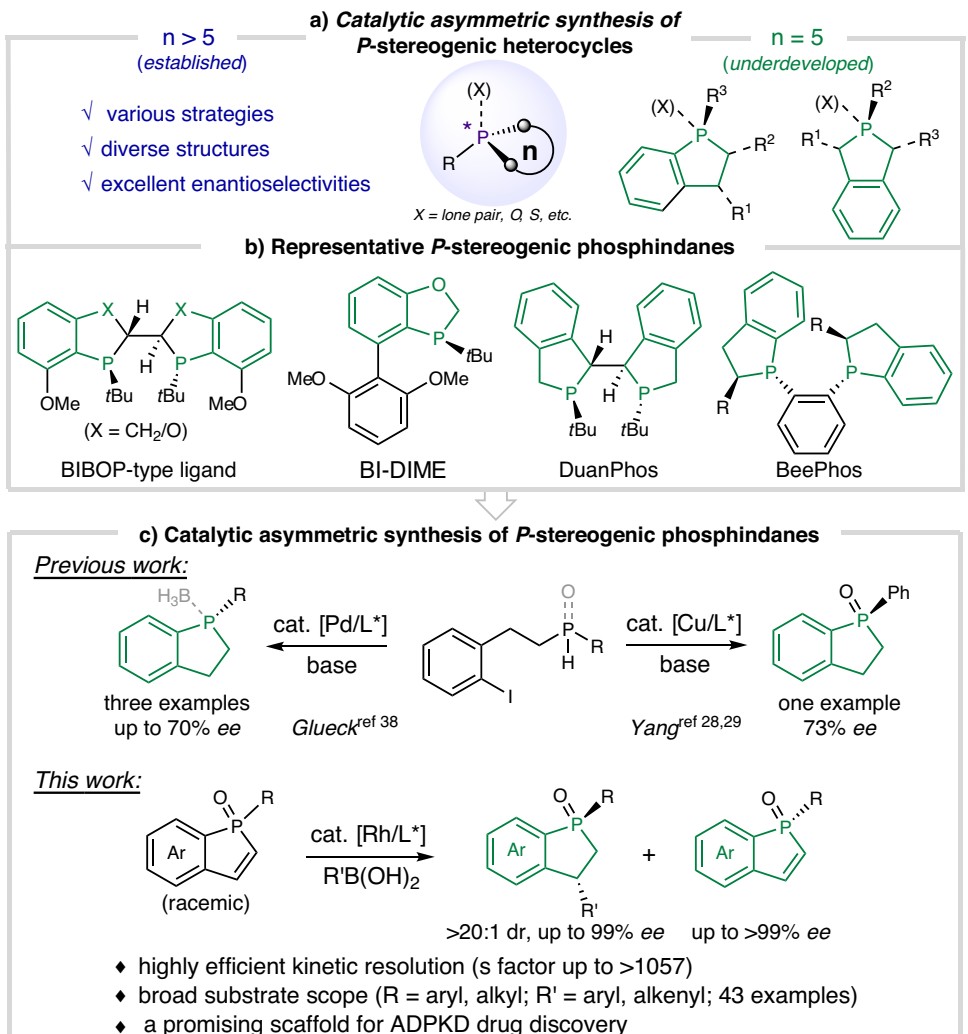

**Fig. 1 | P-Stereogenic phosphindane derivatives. a** Catalytic asymmetric synthesis of *P*-stereogenic heterocycles. **b** Representative *P*-stereogenic phosphindane derivatives. **c** Catalytic asymmetric synthesis of *P*-stereogenic phosphindane derivatives.

developed the copper-catalyzed dynamic kinetic intramolecular cyclization of functionalized secondary phosphine oxides and phosphinates, but the 5-membered product was only obtained with 73% *ee* while the method showed excellent enantioselectivities in producing 6-membered rings and larger ones[28,29]. To the best of our knowledge, the highly enantioselective catalytic synthesis of *P*-stereogenic phosphindane derivatives has not been developed to date.

Phosphindole oxides are widely used in developing functional materials and their synthesis has been well studied[39–41]. They may serve as readily available precursors for the synthesis of *P*-stereogenic phosphindane derivatives via a kinetic resolution process, which is a practically useful strategy for accessing challenging chiral scaffold[42–45]. However, kinetic resolution of phosphindole oxides has not been realized probably due to the difficulties in discriminating the *P*-stereocenter that is away from the potential reaction site. As part of our continued interest in asymmetric synthesis via remote stereocontrol under rhodium catalysis[46–48] and inspired by Hayashi's work on rhodium-catalyzed asymmetric synthesis of *P*-stereogenic phospholanes[35], we envisaged that the remote *P*-stereocenter of phosphindole oxides might be effectively discriminated by a suitable chiral rhodium catalyst, which can be employed for kinetic resolution of racemic *P*-stereogenic phosphindole oxides.

Here, we disclose the successful kinetic resolution of phosphindole oxides via rhodium-catalyzed asymmetric conjugate addition

for the asymmetric synthesis of *P*-stereogenic phosphindane oxides and phosphindole oxides. In addition, the *P*-stereogenic phosphindane oxides are found to exhibit promising therapeutic efficacy in autosomal dominant polycystic kidney disease (ADPKD), which discloses phosphindane derivative a promising scaffold in drug development (Fig. 1c).

## Results

### Reaction development

We initiated our investigation by exploring rhodium-catalyzed asymmetric arylation of racemic phosphindole oxide **1a** with phenylboronic acid **2a**, and the results are summarized in Table 1. First, the chiral bisphosphine ligands that worked well in the asymmetric arylation of phospholene oxides were tested[35]. To our surprise, the reaction not only produced the conjugate arylation product **3aa**, but also gave rise to an unexpected α-arylation product **3aa′**, and the enantioselectivity was only moderate (entries 1-3). These results clearly indicate that the phosphindole oxide differs from the phospholene oxide in terms of both reactivity and enantio-face recognition. We surmised that the chiral diene ligands, which distinguish from the bisphosphine ligands electronically and sterically[49,50], may match the phosphindole oxide substrate to achieve the desired kinetic resolution. Indeed, the chiral diene ligands exhibited different chemo- and stereoselectivity from bisphosphines in this reaction. As exemplified by entry 4 of Table 1, the chiral diene **Ph-bod** led to a highly regioselective generation of the

**Table 1 | Optimization of resolution of phosphindole oxide 1a[a]**

| entry | ligand | 3aa:3aa'[b] | 3aa | | 1a | | C (%)[e] | s[f] |
|---|---|---|---|---|---|---|---|---|
| | | | yield (%)[c] | ee (%)[d] | yield (%)[c] | ee (%)[d] | | |
| 1[g] | (R)-binap | 2.5:1 | 71 | 38 | <5 | — | — | — |
| 2[g] | (R)-segphos | 1.1:1 | 46 | 74 | <5 | — | — | — |
| 3[h] | (R)-segphos | 1.8:1 | 42 | 71 | <5 | — | — | — |
| 4 | (R,R)-Ph-bod | >20:1 | 22 | 73 | 39 | 98 | 57 | 30 |
| 5 | L1 | >20:1 | 31 | 89 | 66 | 45 | 34 | 23 |
| 6 | L2 | >20:1 | 46 | 90 | 48 | 92 | 51 | 53 |
| 7 | L3 | >20:1 | 41 | 94 | 50 | 88 | 48 | 122 |
| 8 | L4 | >20:1 | 49 | 97 | 47 | >99 | 51 | >211 |
| 9 | L5 | >20:1 | 40 | 90 | 44 | >99 | 53 | >80 |
| 10 | L6 | >20:1 | 30 | 81 | 31 | >99 | 55 | >50 |
| 11[i] | L4 | >20:1 | 48 | 94 | 47 | >99 | 52 | >116 |
| 12[j] | L4 | >20:1 | 29 | 76 | 31 | >99 | 57 | >35 |
| 13[k] | L4 | >20:1 | 49 | 97 | 47 | >99 | 51 | >211 |

[a]Unless noted otherwise, the reactions were performed with **1a** (0.20 mmol, 1.0 equiv.), **2a** (1.5 equiv.), catalyst (2 mol%), and KOH (5 mol%) in THF/H$_2$O (1.0/0.1 mL) at 70 °C for 20 h under N$_2$. [b]Determined by $^1$H NMR. [c]Isolated yield. [d]Determined by chiral HPLC. [e]Calculated conversion, $C$ = ee$_{1a}$/(ee$_{1a}$ + ee$_{3aa}$). [f]Selectivity factor (s)=ln[(1-$C$) (1-ee$_{1a}$)]/ln[(1-$C$) (1 + ee$_{1a}$)]. [g]The catalyst was generated in situ from [RhCl(coe)$_2$]$_2$ (2 mol%) and the bisphosphine ligand (5 mol%). [h]1,4-Dioxane instead of THF, [RhCl(coe)$_2$]$_2$ (5 mol % of Rh) and (R)-segphos (10 mol %) was used as the catalyst, and KOH (2.0 mmol) was used. [i]1,4-Dioxane instead of THF. [j]Toluene instead of THF. [k]Conducted at 60 °C. Unless specified otherwise, the dr value of **3aa** was >20:1.

(R)-binap          (R)-segphos          (R,R)-Ph-bod          L1: R = Me / L2: R = Et / L3: R = iPr / L4: R = tBu / L5: R = CH(C$_6$H$_5$)$_2$          L6

conjugate arylation product **3aa** (**3aa:3aa'** > 20:1) with 73% *ee*, and the phosphindole oxide **1a** was recovered with 98% *ee*. Encouraged by this promising result, we examined a series of chiral dienes for this reaction, and the readily available amide-dienes[51,52] proved to be the most effective (entries 5–10). All the examined amide-dienes could maintain the excellent regioselectivity (>20:1), and good selectivity was generally achieved (selectivity factor *s* > 50 for most cases). Among them, the amide-diene **L4** bearing a *N-t*Bu group afforded the best balance of yield and selectivity. Employing **L4** as chiral ligand afforded **3aa** in 49% yield with 97% *ee* and enantiopure **1a** (>99% *ee*) in 47% yield (entry 8, *s* > 211). Other reaction parameters such as the solvent and reaction temperature were also evaluated. The choice of solvent showed to be quite influential on the reaction outcome. For example, the use of 1,4-dioxane only slightly affected the enantioselectivity of **3aa**, but the use of toluene led to a much lower yield and a decreased selectivity (entries 11 and 12). When the reaction temperature was lowered, the enantioselectivity was not improved (entry 13). Thus, the reaction parameters of entry 8 were identified as the optimal conditions for the following studies.

## Substrate scope exploration
With the optimal conditions in hand, we then examined the scope of this catalytic kinetic resolution (Fig. 2). The model reaction produced chiral phosphindane oxide **3aa** (>20:1 dr, 97% *ee*) and enantiopure **1a** (>99% *ee*), and their structures and absolute configurations were determined by single-crystal X-ray diffraction analysis. Arylboronic acids bearing electron-donating substituents (**3ab-3ae**), halogens (**3af-3ai**), and electron-withdrawing substituents (**3aj-3al**) at different positions were all suitable for this transformation, delivering the phosphindane oxides in 40-50% yield with 90-99% *ee*, and the phosphindole oxide **1a** was recovered in 42-50% yield with up to >99% *ee*. Note that a variety of functional groups such as alkenes, halogens and keto carbonyls were tolerated. Aryl-boronic acids featuring a bulky *ortho*-substituent still showed high reactivity and selectivity in this catalytic kinetic resolution system (**3am** and **3an**), and the enantiopure phosphindole oxide (>99% *ee*) could be recovered in 46% isolated yield. Other arylboronic acids, including the multi-substituted (**3ao-3ap**), biphenyl (**3aq**) and polyaromatic (**3ar**) ones also worked well to achieve excellent selectivities (*s* = 152 to >211). Notably, a range of heteroarylboronic acids can participate in the kinetic resolution reaction with a high level of selectivity (*s* >116 in all the examples) to install different heteroarenes, including indole (**3as** and **3at**), quinoline (**3au**), (di)benzofuran (**3av** and **3aw**) and (di)benzothiophene (**3ax** and **3ay**). In the case of 3-benzothienylboronic acid, both the phosphindane oxide product (**3ax**) and the recovered phosphindole oxide (**1a**)

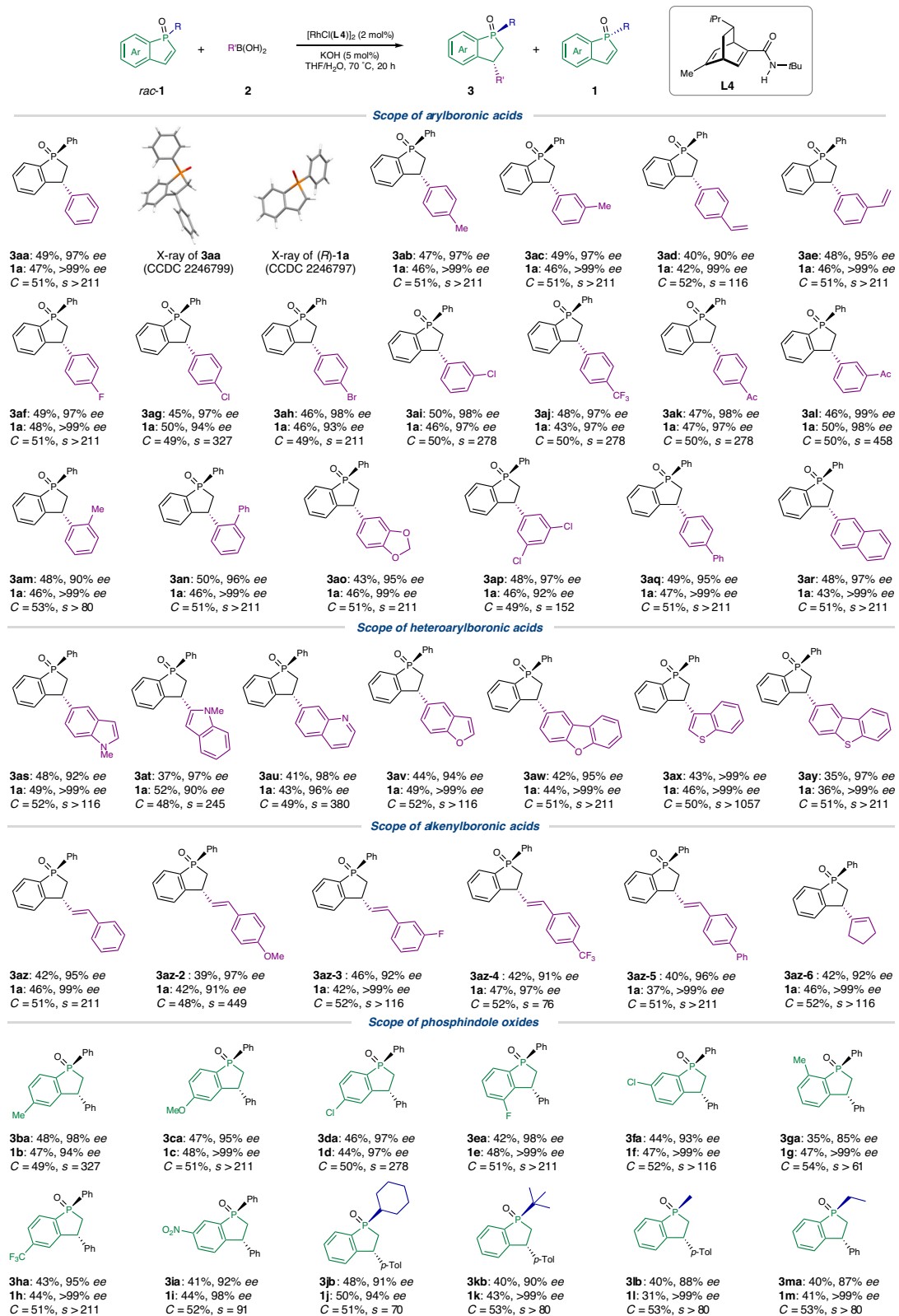

**Fig. 2 | Substrate scope.** Kinetic resolution of phosphindole oxides by Rh-catalyzed asymmetric addition. To a dried Schlenk tube with a magnetic stirring bar were added racemic phosphindole oxide **1** (0.20 mmol), organoboronic acid **2** (0.30 mmol), and [RhCl(L4)]₂ (3.2 mg, 0.004 mmol), followed by the addition of THF (1.0 mL) and aqueous solution of KOH (0.01 mmol in 0.1 mL H₂O) under N₂. The reaction mixture was stirred at 70 °C for 20 h under N₂. Ac acetyl, *p*-Tol *p*-tolyl.

were obtained with >99% *ee*. This selectivity has touched the upper limit of a kinetic resolution process, showcasing the effectiveness of the current protocol. Besides (hetero)arylboronic acids, alkenylboronic acids also underwent the desired kinetic resolution. A variety of styrylboronic acids with different substituents attached to the phenyl ring gave the corresponding products (**3az** to **3az-5**) with high selectivity (*s* > 75). Cyclic alkenylboronic acid was also compatible with the resolution reaction to achieve a high selectivity (**3az-6,** *s* > 116).

The scope of the phosphindole oxides was also explored. When phosphindole oxides bearing diverse substituents at different positions, including 5-position (**3ba-3da**), 4-position (**3ea**), and 6-posistion (**3fa**) were subjected to the catalytic kinetic resolution, similar or even higher levels of reactivity and selectivity can be achieved. The 7-substiutent led to somewhat lower enantioselectivity of the conjugate addition product (**3ga**) probably because of steric hindrance, but the phosphindole oxide was recovered with >99% *ee*. In addition, electron-withdrawing substituents such as $CF_3$ (**3 ha**) and $NO_2$ (**3ia**) were found to be well tolerated. The impact of *P*-substitution groups was also examined. Consistent with the aryl substitution, alkyl groups including primary alkyl groups (Me, Et) and bulky alkyl groups (Cy, *t*Bu) were well tolerated with selectivity factors over 70 (**3jb-3ma**). It is worth highlighting that the *P*-*t*Bu phosphindole oxide (**1k**), which is particularly useful in developing chiral phosphine ligand (Fig. 1b, up), can be recovered in 43% yield with >99% *ee*.

## Synthetic applications

To further demonstrate the practicality of this catalytic system, a 2 mmol-scale kinetic resolution of racemic **1a** was carried out, and chiral phosphindane oxide **3aa** and enantio-enriched phosphindole oxide **1a** could be obtained with the similar efficiency and selectivity as the small scale (Fig. 3a). The chiral phosphine oxides can be readily reduced, thereby providing the valuable chiral phosphines. As shown in Fig. 3b, reduction of **3aa** furnished chiral phosphindane **4** in 96% yield without affecting the diastereo- and enantioselectivity. Double bond was tolerated under the conditions, thus

reduction of **3az** produced phosphine **5** bearing the olefin moiety, which can serve as a good handle for further derivatization or development of phosphine-olefin hybrid ligand. The enantiopure phosphindole oxide provides a good platform for synthesis of chiral phosphindane derivatives. For example, hydrogenation of (*R*)-**1a** afforded the enantiopure phosphindane oxide **6**. Moreover, addition of other nucleophiles to (*R*)-**1a** to generate substituted phosphindane oxide was feasible, as exemplified by the highly diastereoselective addition of diphenylphosphine oxide to generate the enantiopure bisphosphine oxide **7**.

## Mechanistic investigations

Several mechanistic experiments were carried out to gain preliminary insights into the reaction mechanism. First, (*R*)-**1a** was applied in the standard kinetic resolution conditions, and only trace amount of the conjugate addition product could be detected (Fig. 4a). This result indicated that the chiral rhodium catalyst can effectively discriminate the two enantiomers of racemic **1a**, thus rendering the high yield and enantioselectivity of the recovered phosphindole oxide. Next, a deuterium labeling experiment was conducted, and the deuteration at the α position likely arose from direct protonolysis of the alkylrhodium species that was generated by the carborhodation step (Fig. 4b). Finally, the control experiment using *N*-Me-**L4** as the chiral ligand led to a lower yield and a significantly diminished enantioselectivity (Fig. 4c). This result highlights the importance of the secondary amide moiety in enhancing reactivity and enantioselectivity in this arylation reaction, probably via a H-bonding interaction between the amide-NH and substrate. A plausible mechanism of the rhodium-catalyzed kinetic resolution was proposed based on the experimental results and literature report[53,54]. Transmetalation between the active hydroxorhodium catalyst and phenylboronic acids generates the phenylrhodium species, which selectively recognizes (*S*)-**1a** to undergo the subsequent carborhodation step. The transition state is stabilized by hydrogen bonding interactions between the ligand and the substrate (**TS**_{SR}). Finally, hydrolysis of the alkylrhodium species releases the conjugate addition product and regenerates the hydroxorhodium catalyst (Fig. 4d). DFT calculation on the carborhodation step, which is

### a) Scale-up kinetic resolution of *rac*-**1a**

*rac*-**1a** (2.0 mmol)   **2a**
PhB(OH)₂

[RhCl(**L4**)]₂ (2 mol%)
KOH (5 mol%)
THF/H₂O, 70 °C, 20 h

**3aa**
47%, >20:1 dr, 94% *ee*

(*R*)-**1a**
48%, >99% *ee*

### b) Derivatization of the enantio-enriched products

**3aa**
>20:1 dr, 94% *ee*

HSiCl₃, Et₃N
toluene

**4**
96%, >20:1 dr, 94% *ee*

**3az**
>20:1 dr, 95% *ee*

HSiCl₃, Et₃N
toluene

**5**
98%, >20:1 dr, 95% *ee*

**6**
57%, >99% *ee*

Rh(PPh₃)₃Cl, H₂
CH₂Cl₂

(*R*)-**1a**
>99% *ee*

Ph₂P(O)H
*t*BuONa, toluene

**7**
91%, >20:1 dr, >99% *ee*

**Fig. 3 | Synthetic utility. a** Large scale synthesis. **b** Derivatization of the *P*-stereogenic products.

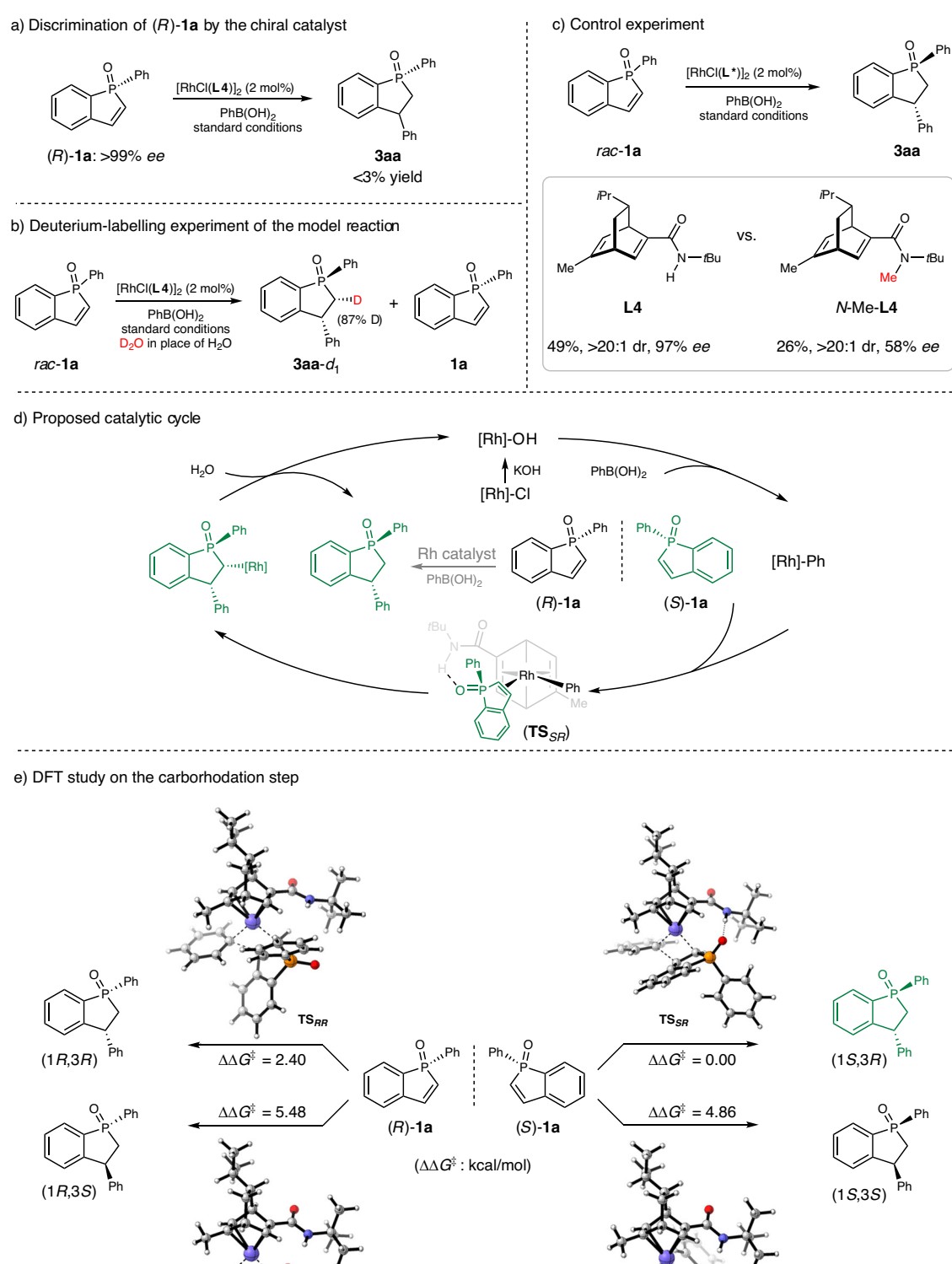

a) Discrimination of (R)-1a by the chiral catalyst

b) Deuterium-labelling experiment of the model reaction

c) Control experiment

d) Proposed catalytic cycle

e) DFT study on the carborhodation step

**Fig. 4 | Mechanistic investigations. a** Discrimination of (R)-**1a** by the chiral catalyst. **b** Deuterium-labeling experiment. **c** Control experiment. **d** Proposed catalytic cycle. **e** DFT study.

decisive for the diastereo- and enantioselectivity of the reaction, was conducted to further understand the observed selectivity. The Gibbs energy differences between competing transition states for the generation of diastereomers are 2.40 kcal/mol (**TS**$_{SR}$ vs **TS**$_{RR}$) and

4.86 kcal/mol (**TS**$_{SR}$ vs **TS**$_{SS}$), which explains the high diastereoselectivity of the reaction. In addition, a Gibbs energy difference of 5.48 kcal/mol accounts for the high enantioselectivity (**TS**$_{SR}$ vs **TS**$_{RS}$). Moreover, the favorable transition state (**TS**$_{SR}$) is found to be stabilized

by hydrogen bonding interactions between the N–H of the ligand and the phosphonyl group of the substrate (see Supplementary Section 2.5 for details).

## Biological activity study

Autosomal dominant polycystic kidney disease (ADPKD) is the most common inherited kidney disorder, and discovery of novel and effective therapeutic agents for this disease is urgently needed[55,56]. Interestingly, 1-indanone was found to retard cyst development in ADPKD[57]. Considering that phosphindane oxide represents a bioisosteric scaffold of 1-indanone, we thus evaluated the therapeutic potential of these chiral phosphindane oxides on ADPKD treatment. Encouragingly, the selected compounds **3aa, 3ay, 3az** and **1a** inhibited Madin-Darby canine kidney (MDCK) cell colony formation, indicating their inhibition on epithelial cell proliferation, which is the most important driving force for renal cyst development in ADPKD, and **3az** displayed better inhibitory efficacy than others (Fig. 5a). We then detected the effect of **3az** on renal cyst development in different ADPKD models. First, we found that **3az** concentration-dependently retarded in vitro MDCK cyst growth (Fig. 5b) and significantly inhibited renal cyst growth in the ex vivo embryonic kidney cyst model (Fig. 5c), which are two widely used models to fast evaluate the pharmacological effect of compounds in ADPKD. Furthermore, **3az** significantly delayed renal cyst growth in vivo in a rapidly progressive ADPKD mouse model (*Pkd1^(flox/flox)^;Ksp-Cre* mice), as indicated by the decreased KW/BW ratio and cyst area in kidneys (Fig. 5d). These data demonstrated that **3az** significantly inhibited renal cyst growth both in vitro and in vivo, and exhibited a potential therapeutic effect in ADPKD. Notably, **3az** showed significantly higher inhibitory effect on renal cyst formation

and development of ADPKD than its enantiomer ***ent*-3az** (see Supplementary Section 2.6 for details). Finally, to initially elucidate the underlying mechanisms of **3az** treatment in ADPKD, we conducted a proteomic analysis of kidney samples from mice with ADPKD and **3az**-treated mice with ADPKD. The result revealed main enrichment in microtubule-based process and microtubule cytoskeleton organization (see Supplementary Section 2.7 for details), which were reported to be relevant to the inhibition of renal cyst expansion and cell proliferation[57]. The docking result also showed that **3az** could stably bind with tubulin (PDB ID: 6EW0, see Supplementary Section 2.8 for details).

## Discussion

In summary, we have realized the highly enantioselective catalytic synthesis of *P*-stereogenic phosphindane derivatives by developing a powerful and practical catalytic system that allows the kinetic resolution of racemic phosphindole oxides in high efficiency and enantioselectivity. The flexibility and practicality of the current catalytic system was demonstrated by the broad scope and the large-scale reaction. Moreover, both experimental and computational investigations were conducted to reveal the reaction mechanism, which involves crucial hydrogen bonding interactions between the ligand and the substrate for high stereocontrol. Finally, the chiral phosphindane oxide products were found to exhibit promising therapeutic efficacy in autosomal dominant polycystic kidney disease, which discloses phosphindane a promising scaffold in drug development. This finding shall help expand the chemical space of drug candidates and inspire the discovery of novel leads for drug development. Further investigations on the utility of *P*-stereogenic phosphindane derivatives in chiral ligand

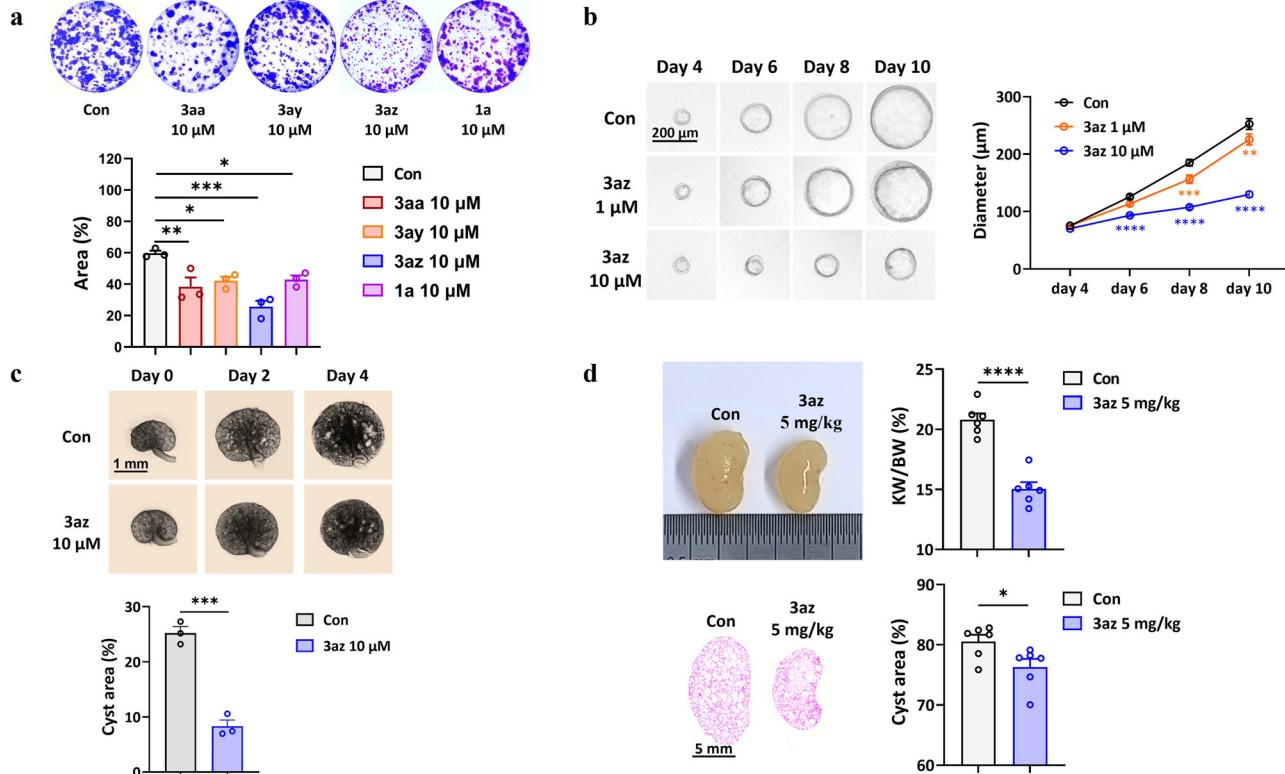

**Fig. 5 | Application of 3az in treatment of ADPKD. a** Inhibition of the MDCK cell colony formation. $n = 3$. $p = 0.0063$, Con vs 3aa; $p = 0.0201$, Con vs 3ay; $p = 0.0002$, Con vs 3az; $p = 0.0265$, Con vs 1a. One-way ANOVA followed by Dunnett's post-test. **b** Concentration-dependent inhibition of the in vitro MDCK cyst growth. $n = 3$. day 6, $p < 0.0001$, Con vs 3az 10 μM. day 8, $p = 0.0004$, Con vs 3az 1 μM; $p < 0.0001$, Con vs 3az 10 μM. day 10, $p = 0.0013$, Con vs 3az 1 μM; $p < 0.0001$, Con vs 3az 10 μM. two-way ANOVA followed by Dunnett's post-test. **c** Inhibition of the renal cyst

growth in the ex vivo embryonic kidney cyst model. $n = 3$. $P = 0.0005$, Con vs 3az 10 μM. two-sided student's $t$ test. **d** Delayed renal cyst growth in vivo in a kidney specific *Pkd1* knockout mouse model. $n = 6$. For KW/BW, $p < 0.0001$. For cyst area, $p = 0.0389$, Con vs 3az. two-sided student's $t$ test. Data are presented as mean ± S.E.M., *$p < 0.05$, **$p < 0.01$, ***$p < 0.001$, ****$p < 0.0001$. Source data are provided as a Source Data file.

development and drug discovery are currently ongoing in our laboratories.

## Methods

### General procedure for the rhodium-catalyzed kinetic resolution of phosphindole oxides

To a dried Schlenk tube with a magnetic stirring bar were added racemic phosphindole oxide **1** (0.20 mmol), organoboronic acid **2** (0.30 mmol), and [RhCl(L4)]$_2$ (3.2 mg, 0.004 mmol), followed by the addition of THF (1.0 mL) and aqueous solution of KOH (0.01 mmol in 0.1 mL H$_2$O) under N$_2$. The reaction mixture was stirred at 70 °C for 20 h under N$_2$. Upon completion, the reaction mixture was diluted with EtOAc (6.0 mL) and water (4.0 mL). The organic layers were separated and the aqueous layer was extracted with EtOAc for two more times (6.0 mL × 2). The combined organic layers were then concentrated in vacuo, and the residue was purified by silica gel chromatography eluting with petroleum ether/THF/EtOH (v/v/ v = 80: 20: 1) to give the phosphindane oxide product **3** and the recovered phosphindole oxide **1**.

### MDCK cell colony formation assay

Madin-Darby canine kidney (MDCK) cells were originally obtained from National Collection of Authenticated Cell Cultures (catalog#GNO23). MDCK cells (approximately 800 cells per well) were seeded in 6-well plates and incubated at 37 °C in 5 % (v/v) CO$_2$ for 24 h. After cell attachment, compounds **3aa, 3ay, 3az,** and **1a** (10 μM) were supplemented to the medium and co-cultured for 8 days. Colony formation assays were performed by fixing cells with paraformaldehyde and stained with 0.5 % crystal violet stain solution (Beyotime Biotechnology, Shanghai, China).

### MDCK cyst model

MDCK cells were originally obtained from National Collection of Authenticated Cell Cultures (catalog#GNO23). MDCK cells were seeded in collagen (PureCol, Inamed Biomaterials, Fremont, CA) and mainly induced by 10 μM forskolin (FSK, #F6886, Sigma, Shanghai, China) to form cysts. During a total incubation period of 10 days, compound **3az** (0, 1, or 10 μM) was added to the culture medium from Days 4 to 10. Micrographs of the cysts (at least 10 cysts/well and 3 wells/group) were taken every two days. Cyst diameters were measured and analyzed using ImageView Software (Nexcope, Ningbo, China).

### Embryonic kidney cyst model

All mice were maintained on a C57BL/6 background. C57BL/6 mice were obtained from the animal facility at Xuzhou Medical University. The kidneys of C57BL/6 mice at embryonic Day 13.5 were cultured on Transwells (Corning 3460, NY, USA). Scattered renal cysts formed and progressively expanded in the presence of 100 μM 8-bromoadenosine 3′,5′-cyclic monophosphate (8-Br-cAMP, #B5386, Sigma, Shanghai, China). Compoud **3az** (10 μM) was added to the culture medium to detect the inhibition of cyst formation and growth. Kidneys were photographed using an inverted microscope (Olympus, Tokyo, Japan). The cyst area and total kidney area were measured using Image-Pro Plus 6.0 software (Rockville, MD, USA).

### ADPKD mouse model

All mice were maintained on a C57BL/6 background. *Pkd1$^{flox/flox}$* mice and *Ksp-Cre* transgenic mice with a C57BL/6 genetic background were obtained from Cyagen (Suzhou, China). To generate *Pkd1$^{flox/-}$; Ksp-Cre* mice we crossed *Pkd1$^{flox/flox}$* and *Ksp-Cre* mice. Neonatal mice were genotyped, and the desired genotype (i.e., *Pkd1$^{flox/flox}$;Ksp-Cre*) were selected at postnatal day 2. All mice used in this study were 12 days old. Both male and female mice were used in autosomal dominant polycystic kidney disease (ADPKD) model. *Pkd1* was specifically knocked out in the kidney to induce the rapid development of renal cysts in the neonatal period. WT (*Pkd1$^{+/+}$;Ksp-Cre*) mice and ADPKD (*Pkd1$^{flox/flox}$;Ksp-Cre*) model mice were both divided into 2 groups: a vehicle control group (10% DMSO, 45% saline, 40% PEG300, and 5% Tween80) and **3az**-treated group (5 mg/kg d$^{-1}$) (*n* = 6). Chemicals were administered to mice by subcutaneous injection on the back every 24 hours from postnatal Day 6 (P6) to P12. The kidneys were harvested and weighed to obtain the kidney index, that is, the ratio of the total kidney weight to body weight (KW/BW). The cyst area and total kidney area were measured using Image-Pro Plus 6.0 software (Rockville, MD, USA). Animal experiments were approved and conducted in accordance with the Laboratory Animal Ethics Committee of Xuzhou Medical University (202309T012).

### Reporting summary

Further information on research design is available in the Nature Portfolio Reporting Summary linked to this article.

## Data availability

All data generated and analyzed in this study can be found in the article and its Supplementary Information. The X-ray crystallographic coordinates for structures reported in this study have been deposited at the Cambridge Crystallographic Data Centre (CCDC), under deposition numbers CCDC 2246799 (**3aa**), CCDC 2246797 ((*R*)-**1a**), and CCDC 2246811 (**3ak**). These data can be obtained free of charge from The Cambridge Crystallographic Data Centre via www.ccdc.cam.ac.uk/ structures/. The crystal structure of tubulin [PDB ID: 6EW0, https:// www.rcsb.org/structure/6EW0] was obtained from Protein Data Bank. The mass spectrometry proteomics data have been deposited to the ProteomeXchange Consortium via the PRIDE partner repository with the dataset identifier PXD049792. The Supplementary Fig. 13a that support the findings of this study is available from BioRender, and was used under licence (agreement number: LU26HODQYI) for this study. The source data underlying Fig. 5, Supplementary Figs. 10-12 and Cartesian Coordinates are provided as a source data file. Data related to materials and methods, optimization of conditions, experimental procedures, mechanistic experiments, and spectra are provided in the Supplementary Information. All data are available from the corresponding authors upon request. Source data are provided with this paper.

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

## Acknowledgements

L.Y. gratefully acknowledges the generous financial support of the National Natural Science Foundation of China (Grant No. 22207094), the Natural Science Foundation of the Jiangsu Higher Education Institutions of China (Grant No. 22KJB350003), and the Youth Foundation of Xuzhou Medical University (Grant No. D2020051). D.G. gratefully acknowledges the generous financial support of the National Natural Science Foundation of China (Grant No. 22077110, 22377103). X.D. is grateful for the generous financial support from the Natural Science Foundation of Jiangsu Province (Grant No. BK20200080) and the opening fund of Hubei Key Laboratory of Bioinorganic Chemistry & Materia Medica (Grant No. BCMM202304). We thank Prof. Hon Wai Lam (University of Nottingham) for the generous donation of the chiral diene ligand L6.

## Author contributions

D.G. and X.D. designed and directed the project. L.Y. supervised the overall execution of the project and carried out experimental procedures with J.L., H.Z., W.Z., Z.F., and M.L. C.W. performed the theoretical calculations and analyzed data. M.G. provided proteomic analysis techniques. X.D. and L.Y. wrote the manuscript. D.G., X.D., and S.Z. made manuscript revisions.

## Competing interests

The authors declare no competing interests.
