## [Peer Review File · Nature Communications]

Asymmetric Synthesis of P-Stereogenic Phosphindane Oxides via Kinetic Resolution and Their Biological ActivityREVIEWER COMMENTS

Reviewer #1 (Remarks to the Author):

This manuscript submitted by Guo and Dou reported the catalytic synthesis method of P-stereogenic benzophospholane oxides via kinetic resolution with their biological activity. While Rhodium-catalyzed asymmetric arylation of phospholene oxides has been reported, the potential ability of Rhodium catalyst with a diverse range of phosphorus compounds is still underdeveloped, especially for producing P-stereogenic compounds. The work is further promoted by DFT study and biological activity analysis, making it a significant contribution to this field. Therefore, this manuscript is recommended for publication in Nat Commun. after addressing the following points:

1. In table 1, the author mentioned that “the chiral diene ligands exhibited different chemo- and stereoselectivity from bisphosphines in this reaction.” Any reason for the good chemo-selectivity of 3aa:3aa' when diene ligand was used?
2. In figure 2, the scope of benzophosphole oxides was limited to the bulky groups (Phenyl, -Cy and -tBu group), how about other examples (e.g.: -Me or OMe)?
3. The author chose ADPKD (Autosomal Dominant Polycystic Kidney Disease) as the potential application for these chiral benzophospholane oxides. Is there any research evidence of these compounds being used for the treatment of this disease?
4. In Figure 5b, the concentration selection for compound 3az is 1 μ M and 10 μ M, respectively. What is the rationale behind choosing the concentration?

Reviewer #2 (Remarks to the Author):

The asymmetric synthesis of P-chiral compound is currently receiving great attention. The present manuscript describes a rhodium-catalyzed kinetic resolution method for production of useful phosphindole derivatives with high enantioselectivity. This work provides a useful KR method to prepare phosphindole derivatives. Moreover, the chiral products were found to exhibit promising therapeutic efficacy in autosomal dominant polycystic kidney disease. Although this type of rhodium-catalyzed arylation was applied previously for DKR of phospholene oxide with high enantioselectivity by the chiral bisphosphine ligands (JACS, 2017, 139, 8122), the chiral bisphosphine ligands cannot achieve high enantioselectivity in the KR of phosphindole derivatives in this manuscript. The authors could obtain high enantioselectivities in KR of phosphindole derivatives with the chiral diene ligand, representing an important advance for this type of transformation. However, before acceptance, corrections are needed.

1. In the last sentence of the first paragraph “To the best of our knowledge, the highly enantioselective synthesis of P-stereogenic benzophospholane derivatives has not been developed to date.” This conclusion is incorrect. Because six-membered and even large membered benzo P-heterocycles have been highly enantioselective asymmetrically synthesized, they are also benzophospholane derivatives.
2. All the reaction substrate and product in this manuscript are phosphindole derivatives, the authors need to give a more accurate description for the substrate and product.

3. "In summary, we have realized the highly enantioselective synthesis of P-stereogenic benzophospholane derivatives for the first time..." This conclusion is incorrect. Re-formulate to make it accurate.

4. This work is inspired by JACS, 2017, 139, 8122. As mentioned by the author "The choice of solvent showed to be quite influential on the reaction outcome". In table 1 entry 1-3, if the reaction condition (the dosage of catalyst and ligand, especially solvent) are the same as JACS, 2017, 139, 8122. Will the regioselectivity and stereoselectivity be improved?

5. In the Scope of benzophosphole oxides, if electron-withdrawing group such as CF₃, NO₂, and a general primary alkyl group replace cyclohexyl group can also work well?

6. In the sentence "Note that a variety of functional groups such as alkenes, halogens and ketones were tolerated." Please change "ketones" to "ketal".

7. The author should introduce the background of relevant medicinal chemistry research in the manuscript.

8. In the bioactivity section, the chiral compound 3az displayed better inhibitory efficacy than others. However, what about the activity of the enantiomer of the compound 3az. Is that configuration is essential for biological activity and binding with tubulin?

9. The ¹³C NMR spectrum of compound 3ai and ¹H NMR spectrum 3am are not pure enough.

Reviewer #3 (Remarks to the Author):

This manuscript reports an impressive kinetic resolution approach to P-stereogenic benzophospholane oxides with remarkably high selectivity. I recommend publication after minor revisions to address some comments about the presentation.

Abstract line 3, "unmet challenge." As explained more clearly in the manuscript, this catalytic process has been done previously. The new and valuable feature here is the higher selectivity and broader scope than in previous work, but this sentence as written is potentially misleading.

Page 2. Why was the crystallographer Rheingold mentioned here?

Page 2, paragraph 1, last sentence, I think this should say highly enantioselective catalytic synthesis, since ref 34 (Stephan) did prepare these compounds by asymmetric synthesis in high ee, with similar wording on top of page 3 ("this method...")

The scope with P-Cy and P-t-Bu groups, as well as the derivatization in Fig 3, is potentially useful.

Mechanistic investigations paragraph: the sentence "Next, the cis..." was confusing. I think it should read "likely arose from..." alkylrhodium species that was generated"

In fig 4, it is hard to see the proposed H-bonding interactions. Could the authors include a Chemdraw diagram here to show them?

Bioactivity: I am not qualified to assess this part. Was there reason to believe the compounds would show activity in this specific area, or was this just random screening?

Response to Reviewers' Comments

We thank the reviewers for their deep engagement with this work entitled "Asymmetric Synthesis of *P*-Stereogenic Benzophospholane Oxides via Kinetic Resolution and Their Biological Activity" (Manuscript number: NCOMMS-23-53505).

We really appreciate all the insightful and constructive comments from the reviewers, which helped us improve our work. We managed to address all the issues and a point-by-point response is shown below.

Reply to comments by Reviewer 1

(1) This manuscript submitted by Guo and Dou reported the catalytic synthesis method of *P*-stereogenic benzophospholane oxides via kinetic resolution with their biological activity. While Rhodium-catalyzed asymmetric arylation of phospholene oxides has been reported, the potential ability of Rhodium catalyst with a diverse range of phosphorus compounds is still underdeveloped, especially for producing *P*-stereogenic compounds. The work is further promoted by DFT study and biological activity analysis, making it a significant contribution to this field. Therefore, this manuscript is recommended for publication in *Nat Commun.* after addressing the following points.

We thank the reviewer for the kind and positive comments.

(2) In table 1, the author mentioned that "the chiral diene ligands exhibited different chemo- and stereoselectivity from bisphosphines in this reaction." Any reason for the good chemo-selectivity of 3aa:3aa' when diene ligand was used?

We thank the reviewer's insightful discussion. The diene ligands are known to be different from the bisphosphine ligands electronically and sterically, thus can lead to different reactivity and selectivity (*Chem. Rev.* **2022**, *122*, 14346.). As a closely related example, switching of chemo-selectivity in Rh-catalyzed arylation using diene/bisphosphine was reported (*J. Am. Chem. Soc.* **2006**, *128*, 5628.). The description and references are added in the revised manuscript (Page 4, paragraph 1).

(3) In figure 2, the scope of benzophosphole oxides was limited to the bulky groups (Phenyl, -Cy and -tBu group), how about other examples (e.g.: -Me or OMe)?

We thank the reviewer's constructive comment. We follow the reviewer's kind suggestion and have tested the benzophosphole oxides bearing primary alkyl groups (Me for **1l**; Et for **1m**) in the current kinetic resolution system. Good selectivity factors ($s > 80$) could still be obtained. The newly added results are included in the revised manuscript (Page 6, Fig 2).

31b: 40%, 88% ee
11: 31%, >99% ee
C = 53%, s > 80

3ma: 40%, 87% ee
1m: 41%, >99% ee
C = 53%, s > 80

(4) The author chose ADPKD (Autosomal Dominant Polycystic Kidney Disease) as the potential application for these chiral benzophospholane oxides. Is there any research evidence of these compounds being used for the treatment of this disease?

We are grateful for this reviewer's helpful comment. Our group has been devoted to the development of novel therapeutic agents for ADPKD treatment (*J. Med. Chem.* **2022**, *65*, 7717; *J. Med. Chem.* **2022**, *65*, 9295; *J. Med. Chem.* **2022**, *65*, 15770; *J. Med. Chem.* **2023**, *66*, 1454; *J. Med. Chem.* **2023**, *66*, 3621.). We noticed that a recent study by Yang and colleagues found that 1-indanone could retard cyst development in ADPKD (*Acta Pharmacol. Sin.* **2023**, *44*, 406.). Considering that benzophospholane oxide represents a bioisosteric scaffold of 1-indanone, we thus evaluated the therapeutic potential of these chiral benzophospholane oxides on ADPKD treatment. The background of medicinal chemistry research on ADPKD is added in the revised manuscript (Page 10, paragraph 1).

(5) In Figure 5b, the concentration selection for compound **3az** is 1 μ M and 10 μ M, respectively. What is the rationale behind choosing the concentration?

We thank the reviewer's careful reading. In fact, 1 μ M and 10 μ M are common starting concentrations for screening compounds with potential therapeutic activity against ADPKD (*J. Am. Soc. Nephrol.* **2019**, *30*, 228; *J. Med. Chem.* **2022**, *65*, 9295; *J. Med. Chem.* **2023**, *66*, 3621.). We have adopted the previously-reported experimental protocol to examine the pharmacological effect of **3az** (SI section 6, page 54).

Reply to comments by Reviewer 2

(1) The asymmetric synthesis of *P*-chiral compound is currently receiving great attention. The present manuscript describes a rhodium-catalyzed kinetic resolution method for production of useful phosphindole derivatives with high enantioselectivity. This work provides a useful KR method to prepare phosphindole derivatives. Moreover, the chiral products were found to exhibit promising therapeutic efficacy in autosomal dominant polycystic kidney disease.

Although this type of rhodium-catalyzed arylation was applied previously for DKR of phospholene oxide with high enantioselectivity by the chiral bisphosphine ligands

(JACS, 2017, 139, 8122), the chiral bisphosphine ligands cannot achieve high enantioselectivity in the KR of phosphindole derivatives in this manuscript. The authors could obtain high enantioselectivities in KR of phosphindole derivatives with the chiral diene ligand, representing an important advance for this type of transformation.

We thank the reviewer for this positive appraisal of our work.

(2) In the last sentence of the first paragraph “To the best of our knowledge, the highly enantioselective synthesis of *P*-stereogenic benzophospholane derivatives has not been developed to date.” This conclusion is incorrect. Because six-membered and even large membered benzo *P*-heterocycles have been highly enantioselective asymmetrically synthesized, they are also benzophospholane derivatives.

We thank the reviewer’s expert comment. We have rectified the description of “benzophospholane” into “phosphindane” as per the reviewer’s suggestion.

(3) all the reaction substrate and product in this manuscript are phosphindole derivatives, the authors need to give a more accurate description for the substrate and product.

We thank the reviewer’s careful reading and the expert input. We have rectified the description of “benzophosphole” to “phosphindole” and “benzophospholane” to “phosphindane” throughout the manuscript.

(4) “In summary, we have realized the highly enantioselective synthesis of *P*-stereogenic benzophospholane derivatives for the first time...” This conclusion is incorrect. Re-formulate to make it accurate.

We thank the reviewer’s careful reading. We have toned down our sentence and rectified the description of “benzophospholane” into “phosphindane”.

(5) This work is inspired by JACS, 2017,139, 8122. As mentioned by the author “The choice of solvent showed to be quite influential on the reaction outcome”. In table 1 entry1-3, if the reaction condition (the dosage of catalyst and ligand, especially solvent) are the same as JACS, 2017,139, 8122. Will the regioselectivity and stereoselectivity be improved?

As per the kind suggestion of the reviewer, we have tried the same reaction conditions as *J. Am. Chem. Soc.* **2017**, 139, 8122. But the regioselectivity and stereoselectivity was not improved (**3aa** : **3aa'** = 1.8:1, yield of **3aa** = 42%, *ee* of **3aa** = 71%). This new result is added in the revised manuscript (Page 4, Table 1, entry 3).

(6) In the Scope of benzophosphole oxides, if electron-withdrawing group such as CF₃, NO₂, and a general primary alkyl group replace cyclohexyl group can also work well? We thank the reviewer for this valuable suggestion. We have tested the phosphindole oxides bearing electron-withdrawing group (CF₃ for **1h**, NO₂ for **1i**), and high selectivity factors comparable to those obtained in the electron-donating group substituted substrates were attainable. In addition, phosphindole oxides bearing *P*-primary alkyl groups (Me for **1l**; Et for **1m**) worked well in the current kinetic resolution system, and good selectivity factors (*s* > 80) could still be obtained. These new results are added in the revised manuscript (Page 6, Fig 2).

3ha: 43%, 95% ee
1h: 44%, >99% ee
C = 51%, *s* > 211

3ia: 41%, 92% ee
1i: 44%, 98% ee
C = 52%, *s* = 91

3lb: 40%, 88% ee
1l: 31%, >99% ee
C = 53%, *s* > 80

3ma: 40%, 87% ee
1m: 41%, >99% ee
C = 53%, *s* > 80

(7) In the sentence “Note that a variety of functional groups such as alkenes, halogens and ketones were tolerated.” Please change “ketones” to “ketal”.

We thank the reviewer’s correction. There is an acyl group instead of a ketal group present in the products, so we have rectified “ketones” to “keto carbonyls” (Page 5, paragraph 2).

(8) The author should introduce the background of relevant medicinal chemistry research in the manuscript.

As per the reviewer’s kind suggestion, we have included the background of relevant medicinal chemistry research as follows (Page 10, paragraph 1).

“...Autosomal dominant polycystic kidney disease (ADPKD) is the most common inherited kidney disorder, and discovery of novel and effective therapeutic agents for this disease is urgently needed. Interestingly, 1-indanone was found to retard cyst development in ADPKD (Acta Pharmacol. Sin. 2023, 44, 406.). Considering that phosphindane oxide represents a bioisosteric scaffold of 1-indanone, we thus evaluated the therapeutic potential of these chiral phosphindane oxides on ADPKD treatment...”

(9) In the bioactivity section, the chiral compound **3az** displayed better inhibitory efficacy than others. However, what about the activity of the enantiomer of the compound **3az**. Is that configuration is essential for biological activity and binding with tubulin?

We thank this insightful comment. We have synthesized the enantiomer of **3az** and tested its biological activities. Compared to compound **3az**, the enantiomer (i.e., *ent*-**3az**) showed significantly decreased inhibitory effects on renal cyst formation and development in *in vitro*, *ex vivo*, and *in vivo* models of ADPKD, demonstrating the importance of configuration on the biological activity. This result is included in the supporting information (SI pages 54-56) and the relevant description is included in the revised manuscript on Page 10.

(10) The ¹³C NMR spectrum of compound **3ai** and ¹H NMR spectrum of **3am** are not pure enough.

We thank the reviewer's critical reading. The ¹³C NMR spectrum of compound **3ai** and ¹H NMR spectrum of **3am** are updated in the Supporting Information (SI section 11, page 80 and page 86).

Reply to comments by Reviewer 3

(1) This manuscript reports an impressive kinetic resolution approach to P-stereogenic benzophospholane oxides with remarkably high selectivity. I recommend publication after minor revisions to address some comments about the presentation.

We thank the reviewer for this positive assessment of our work.

(2) Abstract line 3, "unmet challenge." As explained more clearly in the manuscript, this catalytic process has been done previously. The new and valuable feature here is the higher selectivity and broader scope than in previous work, but this sentence as written is potentially misleading.

Thanks for this valuable comment. We have toned down our sentence by changing the description of "unmet challenge" to "challenging task" (Abstract, line 3).

(3) Page 2. Why was the crystallographer Rheingold mentioned here?

We apologize for this confusion. Rheingold has been removed in the revised manuscript.

(4) Page 2, paragraph 1, last sentence, I think this should say highly enantioselective catalytic synthesis, since ref 34 (Stephan) did prepare these compounds by asymmetric synthesis in high ee, with similar wording on top of page 3 ("this method...") The scope with *P*-Cy and *P*-t-Bu groups, as well as the derivatization in Fig 3, is potentially useful.

We thank the reviewer's valuable input. We have rectified the description accordingly (Page 2, paragraph 1 and Page 3, paragraph 1).

(5) Mechanistic investigations paragraph: the sentence "Next, the cis..." was confusing.

I think it should read “likely arose from”...”alkylrhodium species that was generated”
We follow the reviewer’s suggestion. The sentence was rectified to “Next, a deuterium labeling experiment was conducted, and the deuteration at the α position likely arose from direct protonolysis of the alkylrhodium species that was generated by the carborhodation step (Fig. 4b)” (Page 8, paragraph 2).

(6) In fig 4, it is hard to see the proposed H-bonding interactions. Could the authors include a Chemdraw diagram here to show them?

We thank this important suggestion. The H-bonding interaction was only present in the transition state of TR_{SR} . A Chemdraw diagram of TR_{SR} is include in the proposed catalytic cycle to show the H-bonding interaction (Page 9, figure 4d).

(7) Bioactivity: I am not qualified to assess this part. Was there reason to believe the compounds would show activity in this specific area, or was this just random screening?
Our group has been devoted to the development of novel therapeutic agents for ADPKD treatment (*J. Med. Chem.* **2022**, *65*, 7717; *J. Med. Chem.* **2022**, *65*, 9295; *J. Med. Chem.* **2022**, *65*, 15770; *J. Med. Chem.* **2023**, *66*, 1454; *J. Med. Chem.* **2023**, *66*, 3621.). Interestingly, 1-indanone was found to retard cyst development in ADPKD (*Acta Pharmacol. Sin.* **2023**, *44*, 406.). Considering that phosphindane oxide represents a bioisosteric scaffold of 1-indanone, we thus evaluated the therapeutic potential of these chiral phosphindane oxides on ADPKD treatment. The background of medicinal chemistry research on ADPKD is added in the revised manuscript (Page 10, paragraph 1).

REVIEWERS' COMMENTS

Reviewer #1 (Remarks to the Author):

The authors have addressed most of the concerns raised by the reviewers. It is now recommended to be published in nature communications.

Reviewer #2 (Remarks to the Author):

I am satisfied with the revised manuscript NCOMMS-23-53505A and recommend it to be published in the Nature Communications.

Reviewer #3 (Remarks to the Author):

The revised version has addressed my concerns and is now suitable for publication.